# Critical Tokens Matter:
# Token-Level Contrastive Estimation Enhances LLM's Reasoning Capability

Zicheng Lin [1] [*]   Tian Liang [2] [*]   Jiahao Xu [2] [*]   Qiuzhi Liu [2]   Xing Wang [2]
Ruilin Luo [1]   Chufan Shi [1]   Siheng Li [1]   Yujiu Yang [1]   Zhaopeng Tu [2]

## Abstract

Mathematical reasoning tasks pose significant challenges for large language models (LLMs) because they require precise logical deduction and sequence analysis. In this work, we introduce the concept of **critical tokens** – elements within reasoning trajectories that significantly influence incorrect outcomes. We present a novel framework for identifying these tokens through rollout sampling and demonstrate their substantial divergence from traditional error tokens. Through extensive experiments on datasets such as GSM8K and MATH500, we show that identifying and replacing critical tokens significantly improves model accuracy. We propose an efficient methodology for pinpointing these tokens in large-scale datasets using contrastive estimation and extend this framework to enhance model training processes with direct preference optimization (DPO). Experimental results on GSM8K and MATH500 benchmarks with the widely used models Llama-3 (8B and 70B) and Deepseek-math (7B) demonstrate the effectiveness of the proposed approach, *c*DPO. Our results underscore the potential of leveraging critical tokens to reduce errors in reasoning tasks, advancing the development of AI systems capable of robust logical deduction.

## 1. Introduction

In the domain of artificial intelligence, mathematical reasoning tasks are seen as a crucible for evaluating the proficiency of large language models (LLMs) (Cobbe et al., 2021; Hendrycks et al., 2021; Yuan et al., 2023; Ahn et al.,

*Equal contribution [1]Tsinghua University [2]Tencent. Correspondence to: Yujiu Yang <yang.yujiu@sz.tsinghua.edu.cn>, Zhaopeng Tu <zptu@tencent.com>, First Author <linzc23@mails.tsinghua.edu.cn>.

*Proceedings of the 42nd International Conference on Machine Learning*, Vancouver, Canada. PMLR 267, 2025. Copyright 2025 by the author(s).

2024; Yu et al.; Collins et al., 2024). These tasks necessitate logical and sequential deduction to derive solutions, making them challenging for models trained primarily on general language processing. The chain of thought (COT) method (Wei et al., 2022) has significantly improved the reasoning capability of LLMs by employing a series of intermediate reasoning steps, or reasoning trajectories. Prior research has categorized trajectory errors based on modifications required to correct the COT, such as calculator errors, with the objective of identifying avenues for model improvement (Wei et al., 2022; Wang et al., 2023).

Despite these advancements, the token-level discrepancies within mathematical reasoning contexts have not been systematically explored. Our study seeks to bridge this gap by introducing a novel framework for identifying and quantifying the impact of **critical tokens** on model accuracy. We define critical tokens in mathematical reasoning as *crucial components within an incorrect trajectory that significantly alter the final outcome*. We utilize rollout sampling to identify tokens that substantially influence the correctness of reasoning trajectories. Our findings reveal that critical tokens often diverge from human-annotated error tokens, highlighting their unique role in disrupting logical coherence and computational accuracy. By analyzing the characteristics of critical tokens through word type and positional analysis, we provide novel insights into their nature and influence mechanisms. Furthermore, manipulating a single critical token in incorrect trajectories can significantly enhance accuracy, underscoring their pivotal role in error mitigation.

Building on these insights, we illustrate how critical tokens can enhance reasoning capabilities within Direct Preference Optimization (DPO), a commonly used reinforcement learning algorithm. Although DPO proves effective for general tasks, it encounters difficulties in mathematical reasoning because it may reduce the generation likelihood of positive examples due to lexical similarities with negative examples. Our proposed method, *c*DPO, addresses this issue by targeting critical tokens unique to negative examples, thereby improving the model's ability to differentiate between positive and negative instances. *c*DPO involves the efficient identification and penalization of critical tokens predom-

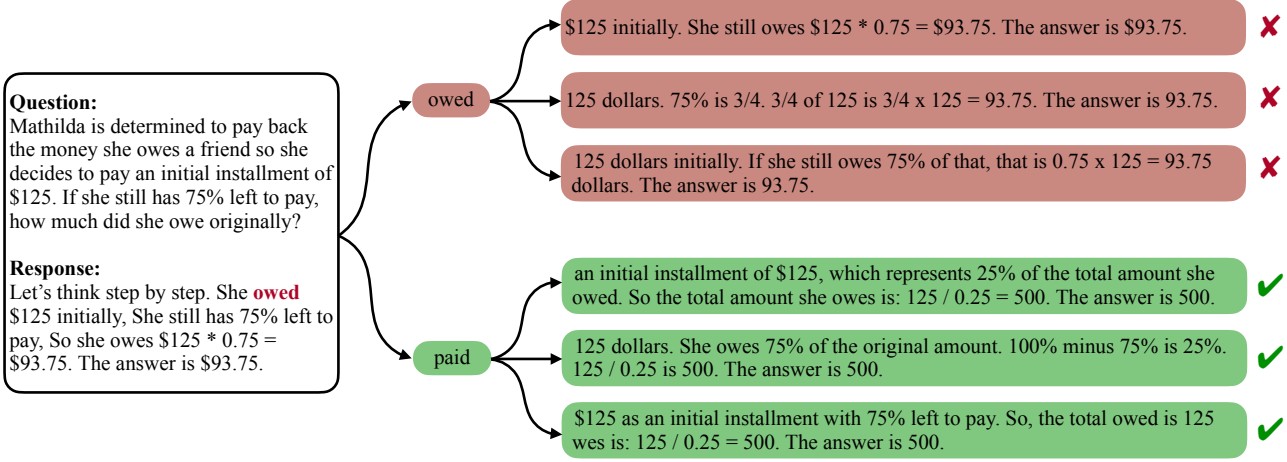

*Figure 1.* An illustration of the critical token "*owed*" shows that it fails to lead to the correct answer in any case. Replacing it with an alternative can significantly increase model accuracy.

inantly found in negative examples, refining the model's learning process and enhancing its understanding of positive outcomes. Experimental results on GSM8K and MATH500 benchmarks using widely recognized models like Llama-3 (8B and 70B) and Deepseek-math (7B) demonstrate that our approach surpasses strong DPO baselines, such as TokenDPO (Zeng et al., 2024) and RPO (Liu et al., 2024; Pang et al., 2024), across all evaluated scenarios.

In summary, our contributions are three-fold:

- We introduce the concept of critical tokens in mathematical reasoning tasks and empirically validate their existence through extensive rollout sampling, distinguishing them from traditional error tokens.

- We propose an efficient approach using contrastive estimation to practically identify critical tokens in large-scale training data, requiring only as little as 0.002% of the computational cost of rollout sampling for GSM8K.

- We develop *c*DPO, an innovative approach that leverages critical tokens within DPO, enhancing the algorithm's ability to distinguish between positive and negative examples in mathematical reasoning.

## 2. Critical Tokens in Mathematic Reasoning

In this section, we explore the concept of "critical tokens" in mathematical reasoning tasks and their impact on model accuracy. We begin by defining critical tokens as pivotal points in incorrect reasoning trajectories that significantly influence outcomes using the example in Figure 1. To validate the presence of critical tokens, we perform rollout sampling, identifying tokens with zero correctness scores that meet

specific conditions in sequence analysis. Our findings reveal that critical tokens differ from human-annotated error tokens in a substantial proportion of cases, emphasizing their unique role in reasoning failure (Table 1). Further experimentation shows that replacing critical tokens boosts model accuracy significantly, highlighting their importance in error reduction (Figure 2). We conclude with an analysis of critical tokens based on word types and positional attributes (Table 2), providing insights into their characteristics.

**Intuition** Mathematical reasoning tasks require logical and sequential deduction to find solutions. We have observed that within incorrect reasoning trajectories, certain tokens are pivotal in leading to incorrect outcomes. These tokens disrupt the logical flow, misrepresent relationships, or introduce computational errors, thus significantly affecting the final result. Unlike other tokens that may have negligible effects on the reasoning process, these "critical tokens" are crucial points of failure. Identifying these tokens is essential because avoiding or correcting them can often result in correct outcomes, even within an incorrect trajectory. As illustrated in Figure 1, the token "owed" is predominantly responsible for incorrect reasoning trajectories as it misguides the logical deduction process. In contrast, prompting the model to decode alternative tokens like "paid" significantly increases the likelihood of producing a correct final result.

**Empirical Validation with Rollout Sampling** To empirically validate the existence of critical tokens, we conducted 64 rollout samplings for each token within an incorrect trajectory. We calculated a score for each token based on the correctness ratio of the generated completions, to quantify its influence on the trajectory. The first token that meets the

*Table 1.* The ratio of incorrect trajectories where critical tokens are different from error tokens across to the error types.

| Error Types | GSM8K Training Data | | MATH500 Training Data | |
|---|---|---|---|---|
| | #Count | Critical != Error | #Count | Critical != Error |
| Calculation Error | 60 | 56.7% | 54 | 79.6% |
| One step Missing | 17 | 72.7% | 8 | 87.5% |
| Semantic Misund. | 22 | 82.3% | 17 | 100.0% |
| Degeneration | 1 | 100.0% | 21 | 95.2% |
| Total | 100 | 65.0% | 100 | 87.0% |

*Table 2.* Analysis of identified critical tokens.

| Error Types | Word Types | | | | | Relative Positions | |
|---|---|---|---|---|---|---|---|
| | Function | Content | Number | Operator | Punct. | Before | After |
| **100 Instances from GSM8K Training Data** | | | | | | | |
| Calculation Error | 10 | 14 | 19 | 10 | 7 | 17 | 17 |
| One step Missing | 5 | 2 | 9 | 0 | 1 | 9 | 5 |
| Semantic Misund. | 3 | 6 | 3 | 3 | 7 | 7 | 9 |
| Degeneration | 0 | 0 | 1 | 0 | 0 | 0 | 1 |
| Total | 18 | 22 | 32 | 13 | 15 | 33 | 32 |
| **100 Instances from MATH500 Training Data** | | | | | | | |
| Calculation Error | 10 | 10 | 15 | 17 | 2 | 30 | 13 |
| One step Missing | 3 | 0 | 2 | 3 | 0 | 6 | 1 |
| Semantic Misund. | 4 | 3 | 2 | 6 | 2 | 10 | 7 |
| Degeneration | 6 | 3 | 6 | 5 | 1 | 10 | 10 |
| Total | 23 | 16 | 25 | 31 | 5 | 56 | 31 |

following two conditions is identified as the critical token:

- The token's correctness score is 0;
- The scores of all subsequent tokens are below 5%.

We analyzed 100 incorrect trajectories from Llama3-8B-base, randomly selected from the MATH training dataset, and successfully identified the critical token in all cases. In addition, by examining 100 random incorrect trajectories from the GSM8K training data, we identified the critical token in 99 cases out of 100. For the outlier case, we identified a critical token that only satisfied the first condition. These results demonstrate the existence of critical tokens.

**Critical Tokens Are *Not* Necessarily Error Tokens** Researchers might hypothesize that critical tokens tend to coincide with error tokens, given their definition (i.e., the correctness ratio of rollout samplings is 0). However, Table 1 demonstrates that critical tokens frequently differ from human-annotated error tokens across various error types (Wei et al., 2022; Wang et al., 2023).

In the GSM8K training dataset, 65% of the critical tokens do not match the error tokens, while this disparity increases

to 87% in the MATH500 training dataset. This variance is further nuanced when examining specific error categories. For example, in the GSM8K data, calculation errors show a 56.7% mismatch, which suggests that nearly half the time the critical tokens identified do not correspond to the actual error tokens. One-step missing error demonstrates a higher 72.7% discrepancy, and semantic misunderstandings show an even greater divergence of 82.3%. Notably, degeneration errors—though based on a single occurrence—exhibit a complete 100% discrepancy with error tokens.

In the MATH500 dataset, a similar pattern is observed. Calculation errors exhibit a 79.6% discrepancy, one-step missing errors an 87.5% mismatch, semantic misunderstandings a 100% divergence, and degeneration errors a significant 95.2% discrepancy. The results in MATH500, particularly the large differences in degeneration errors, underscore the complexity involved in high-precision domains.

These findings underscore a critical insight: while critical tokens are valuable for flagging potential issues in a trajectory, there isn't always a direct correlation with human-annotated error tokens. This divergence emphasizes the intricate nature of error detection and correction in algorithmic analyses,

suggesting that critical tokens capture a broader context of underlying issues, possibly before an error becomes evident.

**Analysis of Critical Tokens** We analyze critical tokens from the following perspectives, as detailed in Table 2.

Tokens are categorized into five types based on their linguistic roles: function words, content words, and numbers. Additionally, operators (e.g., mathematical symbols) and punctuation marks are identified. Our analysis reveals differing patterns across error types and datasets, highlighting the complexity of math reasoning tasks. In the GSM8K dataset, calculation errors predominantly occur with numbers (19 instances) and are fairly distributed among other word types. On the other hand, one step missing errors primarily involve numbers and function words. This suggests an emphasis on numerical manipulation where precision in number usage and function words indicating operations are critical. The MATH500 dataset follows a somewhat similar trend, though it exhibits a higher occurrence of operator-related calculation errors (17 instances), indicating a more frequent use of complex mathematical operations in this dataset. This accentuates the need for careful handling of operators in mathematical computations.

The position of critical tokens is analyzed relative to the corresponding error tokens, categorizing them as occurring before or after the error tokens. If the critical token is the error token itself, it is not counted. In GSM8K, critical tokens are almost evenly distributed: 33 occur before and 32 after the error. However, in MATH500, more critical tokens occur before error tokens, suggesting that critical tokens capture a broader context of underlying issues in the complex MATH500 problems, potentially before an error becomes evident.

**Impact of Critical Tokens** We investigate the effect of critical tokens by replacing them with alternative tokens during model decoding. Specifically, let $t_i$ be a critical token in a given trajectory, and let $T_{<i} = \{t_1, \ldots, t_{i-1}\}$ represent the preceding tokens. We perform $k$ rollout samples based on two different prefixes and calculate the Pass@k metric:

- **w/ critical tokens**: The prefix is $\{t_1, \ldots, t_i\}$. Based on the definition of critical token, all Pass@k scores are zero.

- **w/o critical tokens**: The prefix is $\{t_1, \ldots, t_{i-1}\}$, excluding the critical token $t_i$. We force the model to decode an alternative token at the same position, using the model's probability distribution while masking out the critical token. This substitution allows us to explore if it leads to improved outcomes in model predictions.

Figure 2 displays the results on 100 instances sampled individually from the GSM8K and MATH500 training datasets. By replacing critical tokens with alternative tokens, we

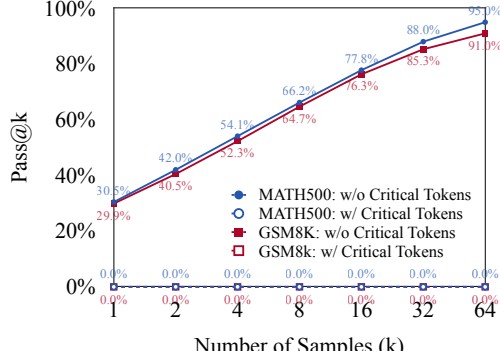

*Figure 2.* Impact of critical tokens on reasoning accuracy. Replacing critical tokens with alternatives ("w/o Critical Tokens") can significantly increase model accuracy on both GSM8K and MATH500, highlighting the importance of these tokens.

observed a significant improvement in the model's performance on these datasets, with Pass@1 accuracy reaching approximately 30% and Pass@64 increasing to over 90%. These results underscore the crucial role that critical tokens play as potential stumbling blocks in the reasoning process. By preventing the model from proceeding with these critical tokens and suggesting alternatives, we enable a higher likelihood of reaching accurate conclusions.

The findings emphasize the importance of understanding and manipulating critical tokens to enhance model performance, especially in complex reasoning tasks. By identifying these critical tokens in reasoning trajectories, we can mitigate the risk of errors, thereby significantly improving the model's effectiveness. Such insights could be instrumental in refining training processes and increasing the reliability of AI systems in real-world applications.

## 3. Enhancing Reasoning Capability with Critical Tokens

In this section, we demonstrate how reasoning capabilities, trained with commonly-used DPO, can be enhanced by using critical tokens. Despite the success of DPO in general instruction tuning tasks, challenges persist when it is applied to reasoning and mathematical tasks. Studies have delved into this issue from an optimization perspective, identifying that these algorithms often diminish the generation likelihood of positive examples in reasoning tasks due to the lexical similarity between positive and negative examples. This overlap can lead to a situation where the model struggles to effectively prioritize and generate the correct trajectory during reasoning or mathematical problem-solving tasks. As a result, the adopted approach is to optimize preferences while ensuring that high generation likelihoods are maintained exclusively for positive examples (Liu et al., 2024; Pang et al., 2024; Pal et al., 2024; Feng et al., 2024).

However, these approaches still attribute high likelihood to tokens that appear in both negative and positive examples, thereby failing to adequately distinguish between truly beneficial features and those that are merely ubiquitous across positive and negative examples.

In this work, we mitigate this problem by leveraging critical tokens that occur only in negative examples. By focusing on these tokens, we aim to enhance the model's ability to effectively differentiate between positive and negative examples. Our approach involves the **identification** and **penalization** of critical tokens that are prevalent exclusively in negative examples. This allows us to adjust the model's learning process by reducing the likelihood of these negative-specific tokens, thereby refining the model's understanding of what constitutes a positive outcome. By explicitly incorporating a mechanism that penalizes frequent negative-example tokens, we ensure that the model learns to prioritize features that truly contribute to successful task completion.

### 3.1. Efficient Identification of Critical Tokens

While it is straightforward to identify critical tokens using rollout sampling as described in Section 2, such methods incur prohibitively high sampling costs and face significant scalability challenges. Moreover, existing methods (Guo et al., 2023; Yoon et al., 2024) depend on external models for token-level annotations, which, although providing effective supervision signals, are costly and limited by the capabilities of the external models. To efficiently identify critical tokens, we propose a method called contrastive estimation, which leverages models trained to learn patterns from both correct and incorrect reasoning trajectories. Figure 3 depicts the framework. By comparing the token-level likelihoods produced by two separately trained models, contrastive estimation can effectively pinpoint critical tokens contributing to incorrect outcomes. The contrastive estimation probability naturally highlights tokens (e.g., "owed") that lead to incorrect reasoning outcomes. We provide additional details throughout the remainder of this section.

**Training Positive and Negative Models**   To implement the contrastive estimation, we need to develop models that can effectively estimate a wide range of both correct and incorrect reasoning distributions. To this end, we collect reasoning trajectories based on the sampling strategy: given a dataset of $M$ instances $\mathcal{D} = \{(x_i, y_i)\}_{i=1}^{M}$, we utilize a pre-trained LLM to sample $N$ reasoning trajectories. Then, we verify the outcome results based on the golden labels $y_i$, which yields $k_i$ positive reasoning trajectories and $N - k_i$ negative reasoning trajectories, which is denoted as:

$$\mathcal{D}^p = \{(x_i, \{y_{i,j}^p\}_{j=1}^{k})\}_{i=1}^{M}$$
$$\mathcal{D}^n = \{(x_i, \{y_{i,j}^n\}_{j=k+1}^{N})\}_{i=1}^{M}$$

For training the positive model, we randomly selected a single correct trajectory because we expect the model to develop decisiveness using its own accurate reasoning paths. For training the negative model, we chose the incorrect trajectories that most frequently occur and account for 50% of all incorrect cases. This approach ensures both variety and representativeness, allowing us to accurately identify critical tokens. For instance, if there are 10 incorrect trajectories comprising 3 cases with the incorrect answer $a$, 2 cases with the incorrect answer $b$, and other cases with answers $\{c, d, e, f, g\}$ occurring only once each, we would randomly select one incorrect trajectory from those with answers $a$ and $b$, as they appear in 5 cases in total. Finally, we train the negative model on this example using two incorrect trajectories: one with answer $a$ and the other with $b$.

**Contrastive Estimation**   With both the positive model and the negative model available, we can automatically annotate the likelihood of each token in an incorrect trajectory being a critical token using contrastive estimation. Let $\mathbf{x}$ be a query, and $\mathbf{y}^n = \{y_1, \ldots, y_t, \ldots, y_T\}$ be a negative example of length $T$ used in DPO training. We compute the likelihood of token $y_t$ being a critical token, denoted as $s_t$, with the following equation:

$$\log s_t = (1 + \beta) \log P^p(y_t | \mathbf{x}, \mathbf{y}_{<t})$$
$$- \beta \log P^n(y_t | \mathbf{x}, \mathbf{y}_{<t}) - \log Z \qquad (1)$$

Here, $\beta$ is a scaling hyperparameter, while $P^p(\cdot)$ and $P^n(\cdot)$ represent the probabilities from the positive and negative models, respectively. The term $\log Z$ is the partition function used in the softmax computation. A low $s_t$ indicates a low likelihood under the correct pattern and a high likelihood under the incorrect pattern, signaling the presence of critical tokens. We further provide theoretical and empirical analysis in Appendix A.

**Efficiency Analysis of Contrastive Estimation**   To evaluate the computational efficiency of contrastive estimation compared to rollout sampling, we estimate the number of forward passes required. Rollout sampling, used to identify critical tokens, incurs substantial inference costs as it relies on sampling from the base model. For GSM8K, obtaining critical tokens through rollout sampling for 100 incorrect examples (64 samples per token) results in an average of 581,425 additional tokens per response (7,613,942 tokens for MATH). Therefore, for $n$ examples, rollout sampling requires approximately $581,425 \times n$ forward passes.

In contrast, contrastive estimation involves both training and inference costs. On GSM8K, the dataset for training the positive and negative models contains 26,131 examples (68,391 examples for MATH), and SFT on this dataset requires approximately $3 \times 26,131 = 78,393$ forward passes, assuming a batch size of 1. For inference on $n$ examples,

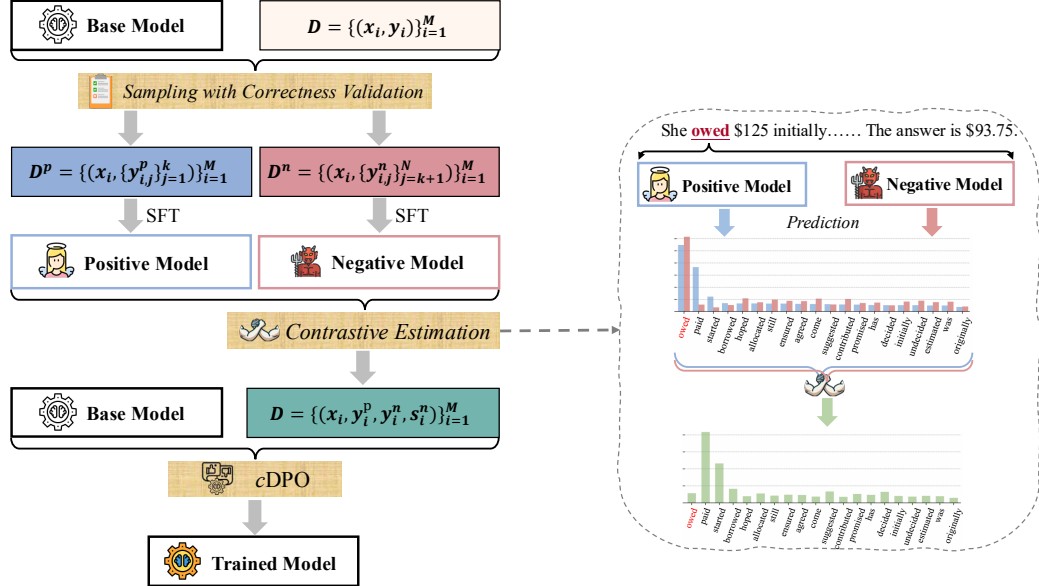

*Figure 3.* Pipeline of the proposed *c*DPO that involves the **efficient identification** and **penalization** of critical tokens that are prevalent exclusively in negative examples.

contrastive estimation only requires each of the positive and negative models to perform one forward pass per example, totaling $2n$. Consequently, the total cost for contrastive estimation is $78,393 + 2n$ forward passes for GSM8K, which is significantly lower than the cost of rollout sampling. For example, for GSM8K that consists of 7,500 examples, contrastive estimation requires only as little as (78393+2*7500)/(581425*7500)=0.002% of the computational cost of rollout sampling. Despite its efficiency, contrastive estimation demonstrates reasonable alignment with rollout-based supervision. Specifically, we measured the Area Under the Curve (AUC) of CE with respect to rollout on LLaMA-3-8B, obtaining values of 0.77 on GSM8K and 0.84 on MATH.

### 3.2. *c*DPO: Explicitly Penalizing Critical Tokens in DPO

**Intuition** Critical tokens in incorrect trajectories significantly contribute to errors, even when other tokens may be correctly placed. By assigning token-level scores to incorrect trajectories, we can specifically penalize critical tokens without adversely affecting correct ones. Conversely, scoring correct trajectories to encourage certain tokens can inadvertently penalize other valid tokens, resulting in undesired distribution shifts in DPO (Rafailov et al., 2024; Xu et al., 2024). Therefore, we focus exclusively on scoring tokens within incorrect trajectories and extend DPO from the example level to the token level by utilizing token-level rewards for preference optimization.

**Formulation** Given the pairwise preference dataset $\mathcal{D} = \{(x_i, y_i^p, y_i^n)\}_{i=1}^M$, the original DPO loss is formulated as:

$$\ell_{\text{DPO}} = -\sum_{i=1}^M \log \sigma(\phi(x_i, y_i^p) - \phi(x_i, y_i^n))$$

Here, $\phi(x, y)$ is an implicit reward function, given by:

$$\phi(x, y) = \gamma \log \frac{\pi_\theta(y \mid x)}{\pi_{\text{ref}}(y \mid x)}$$

where $\pi_\theta(\cdot|x)$ and $\pi_{\text{ref}}(y|x)$ represent the policy model and the reference model, respectively, and $\gamma$ is the coefficient for the KL divergence penalty.

We extend the sample-level DPO to token-level DPO with critical rewards (i.e., *c*DPO). First, we modify the reward function $\phi(x, y)$ to include token-level scores $s_i^n$ (Equation 1) as follows:

$$\phi_s(x, y, s) = \gamma \sum_{t=1}^T (1 - s_t) \log \frac{\pi_\theta(y_t|x, y_{<t})}{\pi_{\text{ref}}(y_t|x, y_{<t})}$$

where $T$ is the total length of the response $y$, and $s_t$ represents the token-level reward score in *c*DPO for the $t$-th token. Accordingly, the objective of *c*DPO is formulated as:

$$\ell_{c\text{DPO}} = -\sum_{i=1}^M \log \sigma(\phi(x_i, y_i^p) - \phi_s(x_i, y_i^n, s_i^n))$$

Note that only the reward function for the negative example $y^n$ is modified. Intuitively, lower values of $s_t$ suggest a

higher likelihood of being critical tokens, which are more prone to result in incorrect outcomes. By weighting each token's contribution with $1 - s_t$, the model effectively penalizes generating these critical tokens. This token-level approach helps ensure that the model reduces the likelihood of generating critical tokens, thus improving the overall accuracy of the responses.

### 3.3. Experimental Results

**Experimental Setup** We used two widely recognized math reasoning datasets: GSM8K (Cobbe et al., 2021) and MATH (Hendrycks et al., 2021). For training, we sampled from all questions in the training set to generate the data. For evaluation, we utilized the MATH500 subset, which is uniformly sampled and has a distribution of difficulty levels and subjects that matches the full MATH test set, as demonstrated in Lightman et al. (2023). Additionally, for both training sampling and evaluation, we adopted few-shot prompting: 8-shot prompting for GSM8K, following Wei et al. (2022), and 4-shot prompting for MATH500, as described in Lewkowycz et al. (2022). For all main evaluations, the temperature was fixed at 0.

We conducted experiments on a range of models, including the general-purpose models Llama-3-8B-base and Llama-3-70B-base (Dubey et al., 2024), as well as the domain-specific model DeepSeek-math-7B-base (Shao et al., 2024). For comparison, we evaluated multiple baseline methods using the data generated from the process described in Section 3.1. For Supervised Fine-Tuning (SFT), we fine-tuned the model using the positive response set $\mathcal{D}^p$. For preference optimization (PO) methods, we utilized the token-level annotated pair-wise preference dataset $\mathcal{D}$. The baselines we compared include:

- **DPO** (Rafailov et al., 2024): We tested two different starting points for training: based on the base model and on the SFT model.

- **TokenDPO** (Zeng et al., 2024), which is a token-level approach that enhances Kullback-Leibler (KL) divergence regulation by incorporating forward KL divergence constraints at the token level. The SFT model is used as the starting point for training. We implemented TDPO using the publicly available implementation.

- **RPO** (Liu et al., 2024; Pang et al., 2024) introduces an extra negative log-likelihood term to improve performance on reasoning tasks. We implemented it using Hugging-Face's implementation and starting with the base model.

We used LoRA adapters (Hu et al., 2022) to train all the models. We trained both positive and negative models for 1 epoch with a learning rate of 3e-4. For preference optimization training, we set $\gamma = 1.0$ and trained for 3 epochs

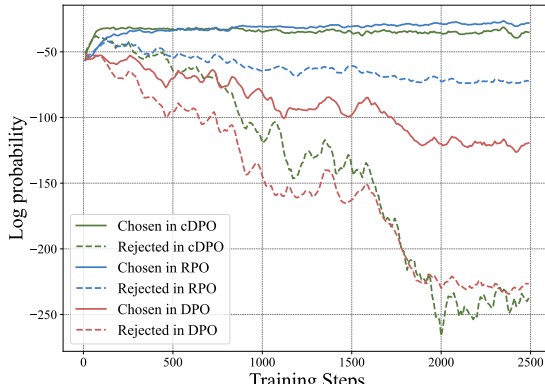

*Figure 4.* Log probabilities of chosen and rejected sequences during training on the GSM8K dataset. Solid lines represent chosen sequences, while dashed lines denote rejected sequences. The figure illustrates how $c$DPO achieves a better separation between chosen and rejected sequences compared to DPO and RPO.

with a learning rate of 2e-5 for all baseline methods. For our $c$DPO approach, since the token-level scores range between 0 and 1 (whereas in DPO, the scores were all 1), we simply increased the learning rate to 4e-5.

For our proposed **$c$DPO**, each problem was sampled $N = 64$ times, selecting the top-$p = 50\%$ of incorrect trajectories to train the negative model $q(\cdot)$. During estimation, the hyperparameter $\beta$ was set to 1.0.

**Learning Curves** We begin by examining the impact of $c$DPO on training dynamics. Figure 4 shows the log probability trends for selected and non-selected sequences over training steps on the GSM8K dataset using the Llama-3-8B model with DPO, RPO, and $c$DPO.

The proposed $c$DPO method successfully differentiates between chosen and rejected sequences by significantly increasing the log probability of correct sequences while sharply decreasing that of incorrect ones. In comparison, RPO (DPO with an additional NLL term) increases the probability of correct sequences, but its impact on reducing incorrect response probabilities is less significant. On the other hand, DPO notably decreases the probability of generating incorrect sequences, but also reduces the probability of correct sequences. This suggests that $c$DPO strikes a balanced approach, effectively enhancing the probability of correct outputs while minimizing critical errors, exceeding the performance of both DPO and RPO.

**Main Results** Table 3 presents the experimental results for various methods across the GSM8K and MATH500. Our proposed method consistently outperforms all baselines and other methods, achieving the highest scores across both datasets. For the GSM8K dataset, our method achieves a

| Method | GSM8K | | | | MATH500 | | | |
| | Llama-3 | | DeepSeek | Avg. | Llama-3 | | DeepSeek | Avg. |
| | 8B | 70B | math-7B | | 8B | 70B | math-7B | |
| Baseline | 56.4 | 80.4 | 64.1 | 67.0 | 18.6 | 43.6 | 34.0 | 32.1 |
| + SFT | 61.2 | 82.1 | 67.1 | 70.1 | 16.8 | 43.4 | 34.2 | 31.5 |
| + DPO | 59.7 | 87.8 | 66.5 | 71.3 | 17.6 | 40.8 | 34.6 | 31.0 |
| + TokenDPO | 62.3 | 83.3 | 69.6 | 71.7 | 19.2 | 43.6 | 34.6 | 32.5 |
| + DPO | 59.6 | 88.9 | 63.1 | 70.5 | 14.6 | 42.6 | 35.0 | 30.7 |
| + RPO | 67.5 | 89.7 | 68.9 | 75.4 | 19.4 | 44.0 | 33.6 | 32.3 |
| + cDPO (Ours) | **67.9*** | **90.8*** | **72.9*** | **77.2*** | **19.6*** | **45.0*** | **35.2*** | **33.3*** |

*Table 3.* Experimental results on GSM8K and MATH500 datasets. Our proposed method surpasses all the strong baselines at a large margin on individual settings and average performance. * denotes the significance test where $p < 0.005$.

remarkable average score of 77.2, surpassing the Baseline and notable improvements such as those incorporating SFT and DPO. Specifically, our approach reaches the highest scores with Llama-3 (90.8 for 70B) and DeepSeek (72.9). These results highlight the effectiveness of our method in leveraging the strengths of both large-scale models (Llama-3) and task-specific models (DeepSeek).

Similarly, on the MATH500 dataset, our method attains an average score of 33.3, marking an improvement over the baseline (32.1) and other enhanced methods such as SFT and RPO. Notably, our approach yields the highest individual score with Llama-3 (45.0 for 70B) and performs robustly across all model configurations.

Since models exhibit sensitivity to prompt formatting when evaluated with temperature 0 on MATH500, we further perform additional analyses by sampling each question 10 times under varying temperatures and report Pass@1 in the Table 5. As shown, cDPO consistently outperforms the corresponding base models across all temperature settings and exhibits stable, robust performance.

To further verify the effectiveness of our method on stronger models, we conduct experiments using Qwen-2.5-7B and Qwen-2.5-32B on both GSM8K and MATH500. The results are presented in Table 6. As shown, cDPO consistently improves performance across both datasets and model scales.

The consistent performance improvements observed across various settings underscore the superiority of our method compared to existing techniques. The significance tests, which were conducted to verify the statistical reliability of these results, confirm the competitive advantage of our proposed approach.

## 4. Related Work

**Contrastive Estimation** Notable works have refined contrastive estimation techniques (Gutmann & Hyvärinen,

2010; Bose et al., 2018; He et al., 2020; Denize et al., 2023). Specifically, our work is closely related to contrastive decoding (CD), an application of contrastive estimation in downstream tasks. CD (Li et al., 2023) involves contrasting token distribution likelihoods between expert and amateur models during decoding. As described by O'Brien & Lewis (2023), this technique avoids high-probability but low-quality tokens, ensuring text fluency and coherence.

Subsequent research has emphasized CD's potential to improve factuality (Zhang et al., 2023; Yang et al., 2024), knowledge editing (Bi et al., 2024), safety (Zhao et al., 2024), and reasoning (O'Brien & Lewis, 2023; Shi et al., 2024; Yuan et al., 2024; Yang et al., 2025). Notably, O'Brien & Lewis (2023) showed that CD enhances reasoning tasks and mitigates typical errors. Shi et al. (2024) demonstrated that unchosen experts in Mixture-of-Experts models could be applied for CD, thereby improving model reasoning capacities. Different from those works that focus on the inference of contrastive decoding, our research primarily uses contrastive estimation to identify "critical tokens" that significantly affect the correctness of the reasoning process.

**Reinforcement Learning** Among various alignment algorithms (Christiano et al., 2017; Schulman et al., 2017; Ziegler et al., 2019; Ouyang et al., 2022; Bai et al., 2022), DPO (Rafailov et al., 2024) is one of the most representative algorithms. DPO uses the LLM itself as a secret reward model and conducts preference optimization on a preference pair of positive and negative examples. Since then, various contributions have been made to further advance the DPO development of LLM alignment (Pal et al., 2024; Amini et al., 2024; Azar et al., 2024).

A line of research aims to improve the performance of DPO in reasoning tasks (Liu et al., 2024; Pang et al., 2024; Pal et al., 2024; Feng et al., 2024). Furthermore, Lai et al. (2024) leverage human or GPT-4 validation to pinpoint incorrect reasoning steps; Guo et al. (2023); Yoon et al. (2024) utilize

external LLMs to refine responses, deriving token-level preferences from pre- and post-revision comparisons. In contrast, our study seeks to establish an automatic process supervision strategy devoid of human annotation and easy to scale. Specifically, we harness contrastive estimation to identify critical tokens, providing token-level signals for preference optimization that significantly enhance LLM reasoning capabilities.

## 5. Conclusion

Our work contributes a significant framework for understanding and enhancing mathematical reasoning in LLMs through critical token analysis. By defining and identifying critical tokens, we provide valuable insights into token-level discrepancies that disrupt logical reasoning. Our approach, *c*DPO, successfully integrates this analysis into DPO, improving model performance on mathematical tasks by improving the model's differentiation between positive and negative examples. Experimental results from GSM8K and MATH500 benchmarks have shown that our method outperforms existing DPO baselines, underscoring the potential of critical token interventions in enhancing model accuracy. Our research opens new doors for further exploration of token-level influences in complex reasoning tasks, which could lead to more refined and effective LLMs.

Future work should explore the integration of *c*DPO with other reasoning frameworks and extend its application to diverse logical reasoning domains, contributing towards the broader aim of developing more robust and reliable LLMs.

## Impact Statement

In our research, we focus exclusively on developing models for solving mathematical problems, which inherently minimizes common ethical concerns typically associated with AI applications in broader domains. The primary function of our models is to enhance computational accuracy and efficiency in mathematical tasks, ensuring that any potential bias, privacy issues, or harmful outputs are effectively nonexistent. Since our dataset consists solely of mathematical questions, it does not involve personal, sensitive, or controversial information that could lead to ethical dilemmas.

## Acknowledgments

This work was partly supported by the National Key Research and Development Program of China (No. 2024YFB2808903) , the Shenzhen Science and Technology Program (JCYJ20220818101014030) and the research fund of Tsinghua University - Tencent Joint Laboratory for Internet Innovation Technology.

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

# A. Contrastive Estimation: Theoretical and Empirical Analysis

## A.1. Distribution Analysis of Contrastive Estimation

We demonstrate that contrastive estimation does not fundamentally alter the nature of the trajectory distribution. According to (Guo et al., 2023; Lambert et al., 2024), the trajectory distribution can be modeled as Gaussian distributions based on correctness. Consequently, we define the probability density functions for the correct and incorrect distributions, denoted as $P^p$ and $P^n$ respectively, as follows:

$$P^p(x) = \frac{1}{\sqrt{2\pi}\sigma} \exp\left(-\frac{(x-\mu_{\mathrm{p}})^2}{2\sigma^2}\right), \tag{2}$$

$$P^n(x) = \frac{1}{\sqrt{2\pi}\sigma} \exp\left(-\frac{(x-\mu_{\mathrm{n}})^2}{2\sigma^2}\right), \tag{3}$$

where the means satisfy $\mu_{\mathrm{p}} > \mu_{\mathrm{n}}$, and both distributions share the same standard deviation $\sigma$, facilitating a straightforward comparison between them.

Therefore, according to Eq. 1, the probability density function $P^{\mathrm{ce}}$ of the CE distribution can be calculated as follows:

$$\log(P^{\mathrm{ce}}(x)) = (1+\beta)\log(P^p(x)) - \beta\log(P^n(x)) - \log(Z_1), \tag{4}$$

where $Z_1$ is the partition function.

Substituting the definitions of $P^p(x)$ and $P^n(x)$, we obtain:

$$\log(P^{\mathrm{ce}}(x)) = \log\left(\frac{1}{\sqrt{2\pi}\sigma}\right) - \frac{(1+\beta)(x-\mu_{\mathrm{p}})^2 - \beta(x-\mu_{\mathrm{n}})^2}{2\sigma^2} - \log(Z_1). \tag{5}$$

Thus, the CE distribution $P^{\mathrm{ce}}$ is:

$$P^{\mathrm{ce}}(x) = \frac{1}{Z_1}\left(\frac{1}{\sqrt{2\pi}\sigma} \exp\left(-\frac{(1+\beta)(x-\mu_{\mathrm{p}})^2 - \beta(x-\mu_{\mathrm{n}})^2}{2\sigma^2}\right)\right). \tag{6}$$

The term $(1+\beta)(x-\mu_{\mathrm{p}})^2 - \beta(x-\mu_{\mathrm{n}})^2$ can be expressed as:

$$(1+\beta)(x-\mu_{\mathrm{p}})^2 - \beta(x-\mu_{\mathrm{n}})^2 = (x-\mu^{\mathrm{ce}})^2 + Z_3, \tag{7}$$

where $\mu^{\mathrm{ce}} = \mu_{\mathrm{p}} + \beta(\mu_{\mathrm{p}} - \mu_{\mathrm{n}})$, and $Z_3$ is a constant independent of $x$.

Substituting this result into (6) gives:

$$P^{\mathrm{ce}}(x) = \frac{1}{Z_1}\left(\frac{1}{\sqrt{2\pi}\sigma} \exp\left(-\frac{(x-\mu^{\mathrm{ce}})^2}{2\sigma^2}\right) \exp\left(-\frac{Z_3}{2\sigma^2}\right)\right), \tag{8}$$

where $Z_1 = \exp\left(-\frac{Z_3}{2\sigma^2}\right)$. Finally, the CE distribution can be written as:

$$P^{\mathrm{ce}}(x) = \frac{1}{\sqrt{2\pi}\sigma} \exp\left(-\frac{(x-\mu^{\mathrm{ce}})^2}{2\sigma^2}\right). \tag{9}$$

Hence, $P^{\mathrm{ce}}(x)$ is also a Gaussian distribution with mean $\mu^{\mathrm{ce}} = \mu_{\mathrm{p}} + \beta(\mu_{\mathrm{p}} - \mu_{\mathrm{n}})$ and variance $\sigma^2$. Here, contrast factor $\beta$ controls the extent to which the distribution shifts away from the negative mean $\mu_{\mathrm{n}}$ towards the positive mean $\mu_{\mathrm{p}}$

## A.2. Effect of the Contrast Factor $\beta$

To better understand the influence of the scaling factor $\beta$ in contrastive estimation, we conduct an ablation study using the LLaMA-3-8B model on the GSM8K. The results are shown in Table 4.

*Table 4.* Ablation study on the contrast factor $\beta$ on GSM8K using LLaMA-3-8B.

| $\beta$ | 0.5 | 0.75 | 1.0 | 1.25 | 1.5 | 1.75 | 2.0 | 2.25 | 2.5 |
|---|---|---|---|---|---|---|---|---|---|
| $c$DPO (%) | 66.5 | 69.2 | 67.9 | 67.2 | 69.5 | **70.7** | 68.4 | 66.8 | 68.9 |

As discussed in Appendix A.1, the scaling factor $\beta$ shifts the mean of the modified distribution $P^{\text{ce}}$ away from the negative mean $\mu_{\text{n}}$ toward the positive mean $\mu_{\text{p}}$, following $\mu^{\text{ce}} = \mu_{\text{p}} + \beta(\mu_{\text{p}} - \mu_{\text{n}})$. The ablation results show that setting $\beta$ in the range of 1.5 to 1.75 leads to better performance, indicating that $P^{\text{ce}}$ aligns more closely with the optimal distribution when the contrast is moderately emphasized.

## B. Futhrer Experiments

### B.1. Pass@1 accuracy on MATH500

*Table 5.* Pass@1 accuracy on MATH500. Each result is averaged over 10 samplings per question.

| Temperature | 0 | 0.25 | 0.5 | 0.75 | 1.0 | 1.25 | 1.5 |
|---|---|---|---|---|---|---|---|
| LLaMA-3-8B | 18.6 | 16.4 | 15.3 | 13.0 | 9.5 | 3.3 | 1.2 |
| + $c$DPO | **19.6** | **20.3** | **20.1** | **19.6** | **18.7** | **19.7** | **18.3** |
| DeepSeek-Math-7B | 34.0 | 31.8 | 30.5 | 26.2 | 21.1 | 11.3 | 3.0 |
| + $c$DPO | **35.2** | **34.5** | **34.5** | **34.5** | **33.9** | **32.9** | **32.8** |
| Qwen-2.5-7B | 49.2 | 46.9 | 45.1 | 41.4 | 34.0 | 20.1 | 2.8 |
| + $c$DPO | **54.0** | **54.2** | **53.6** | **52.8** | **52.9** | **53.4** | **51.9** |

### B.2. Accuracy of Qwen-2.5 models

*Table 6.* Accuracy of Qwen-2.5 models on GSM8K and MATH500.

| Model | GSM8K | MATH500 |
|---|---|---|
| Qwen-2.5-7B | 85.5 | 49.2 |
| + cDPO | **87.5** | **54.0** |
| Qwen-2.5-32B | 93.0 | 58.8 |
| + cDPO | **93.5** | **64.8** |

