# OpenReview forum: "Critical Tokens Matter: Token-Level Contrastive Estimation Enhances LLM’s Reasoning Capability"
_ICML.cc/2025/Conference — ICML 2025 poster_

### Official Review · Reviewer_W2W8 · 2025-03-03

**Overall Recommendation:** 3

**Summary:**

The paper introduces the concept of critical tokens in mathematical reasoning tasks, which are pivotal points in incorrect reasoning trajectories that significantly influence the final outcome. The authors propose a novel framework for identifying these tokens through contrastive estimation and they further introduce cDPO. Reducing the occurrence probability of critic tokens through the DPO is straightforward.

## update after rebuttal
I will keep my positive score because my concerns have been partially addressed.

**Claims And Evidence:**

Yes

**Essential References Not Discussed:**

I am not familiar enough with this area to seriously recommend anything.

**Experimental Designs Or Analyses:**

Yes

**Methods And Evaluation Criteria:**

Yes

**Other Comments Or Suggestions:**

This is discussed in the Strengths And Weaknesses.

**Other Strengths And Weaknesses:**

**Strength**

1. The introduction of critical tokens is a novel and insightful contribution to the field of mathematical reasoning in LLMs. The authors provide a clear definition and empirical validation of these tokens, showing their significant impact on model accuracy.

2. This paper is well-written and easy to follow. The paper provides sufficient technical details for readers to understand.


**Weakness**

1. A model trained with incorrect trajectories is used to simulate the probability distribution of critic tokens, but the incorrect trajectories do not necessarily satisfy the two conditions on page 2.

2. Lack the sensitivity of the proposed method to hyperparameters, such as the scaling factor β in contrastive estimation. Understanding how sensitive the results are to these parameters would be valuable for practitioners looking to implement the method.

**Questions For Authors:**

The Question is discussed in the Weaknesses.

**Relation To Broader Scientific Literature:**

I am not familiar enough with this area to seriously recommend anything.

**Theoretical Claims:**

Yes

---

> ### Author Rebuttal · Authors · 2025-04-01
>
> Thank you very much for your insightful and detailed comments. Below, we address each concern and hope that our responses sufficiently clarify your questions.
>
> **Weakness**
>
>  **W1. Critical Token Estimation**
>
> In our approach, the distribution for critic tokens is approximated using a model trained with incorrect trajectories. Although these incorrect trajectories do not necessarily satisfy the two conditions specified on page 2, our experimental results nevertheless indicate robust and competitive performance.
>
> We measured the Area Under the Curve (AUC) of Contrastive Estimation (CE) with respect to rollout on LLaMA-3-8B, obtaining values of: (1) 0.77 on GSM8K, and (2) 0.84 on MATH. These metrics and corresponding analyses will be explicitly included in our revised manuscript.
>
> **W2. Ablation Study for Hyperparameter Sensitivity**
>
> Thank you for pointing out this important issue. We fully agree that examining hyperparameter sensitivity will strengthen the paper, enhancing its usefulness for practitioners. Accordingly, we have now included an ablation study investigating different values of the scaling factor β using the LLaMA-3-8B model on GSM8K, as summarized in the table below.
>
> | β-value | 0.5  | 0.75 | 1.0  | 1.25 | 1.5  | 1.75 | 2.0  | 2.25 | 2.5  |
> |---------|------|------|------|------|------|------|------|------|------|
> | cDPO    | 66.5 | 69.2 | 67.9 | 67.2 | 69.5 | 70.7 | 68.4 | 66.8 | 68.9 |
>
> As further elaborated in Appendix section 'Distribution Analysis of Contrastive Estimation,' the hyperparameter β affects the mean of the modified distribution P^{ce}. The above ablation results demonstrate that selecting β in a suitable range (approximately 1.5 to 1.75) shifts P^{ce} toward a more accurate distribution, thereby improving overall model performance.

---

### Official Review · Reviewer_KGMH · 2025-03-13

**Overall Recommendation:** 3

**Summary:**

In this paper, the authors start from the observation that the existence of critical tokens will influence the model performance and propose a contrastive estimation method to identify the critical tokens. At last, the authors proposed the cDPO to improve the model performance.

**Claims And Evidence:**

Yes

**Essential References Not Discussed:**

NaN

**Experimental Designs Or Analyses:**

yes

**Methods And Evaluation Criteria:**

Yes

**Other Comments Or Suggestions:**

NaN

**Other Strengths And Weaknesses:**

### Strength
1. The paper is well-written and easy to follow.
2. The observation and analysis are novel and insightful.
3. Results of the proposed methods work well.

### Weakness.
1. My main concern is the correctness of the baseline results. I find the baseline results are lower than those reported in the original paper. For example, in Table 2 of DeepSeekMath, its performance on MATH is 36.2, not 31.4 as reported in the paper. Also, from the Table. 12 in LLaMA3, the LLaMA 3-8B gets 20.3 on MATH, but not 16.8. This is a serious problem, and I think the authors should clarify it in the rebuttal.

2. The experiment is somewhat simple, with only three models and two benchmarks. This also hinders the convincness of the proposed methods.

3. Similar to 2, there are no ablation and few analyses of the proposed methods.

In a word, I think the paper starts from an interesting observation and convincing analyses. But the results of the experiment part have some problems. So, I give a weak reject to the current version, and I'll adjust the final rating based on the rebuttal.

## update after rebuttal
The rebuttal partly solved my concerns. So I raise my score to weak accept.

**Questions For Authors:**

NaN

**Relation To Broader Scientific Literature:**

NaN

**Theoretical Claims:**

NaN

---

> ### Author Rebuttal · Authors · 2025-04-01
>
> Thank you very much for your constructive feedback and your willingness to engage in further discussions with us. We have responded to each of the issues you raised below and have carefully addressed all your concerns.
>
> **Weakness**
>  **W1. Baseline results**
>
> Thank you for pointing out this important issue. Upon careful investigation, we discovered that this disparity stems from a formatting inconsistency in the few-shot prompting examples. Specifically, the prompts differed slightly in spacing:
>   - "Problem:\n" (line break)
>   - versus "Problem: " (single space)
>
> This seemingly minor formatting difference unfortunately resulted in underestimated baseline performance on the MATH500 dataset. We have since corrected this issue. The revised performance numbers are reported below:
>
> | Model               | MATH500 |
> |---------------------|---------|
> | DeepSeek-math-7B    | 34.0    |
> | + cDPO              | 35.2    |
> | Llama-3-8B          | 18.6    |
> | + cDPO              | 19.6    |
> | Llama-3-70B         | 44.4    |
> | + cDPO              | 45.0    |
> | Qwen-2.5-7B         | 49.2    |
> | + cDPO              | 54.0    |
> | Qwen-2.5-32B        | 58.8    |
> | + cDPO              | 64.8    |
>
> We will update and clarify the results accordingly in the revised manuscript.
>
>  **W2. More experiments**
>
> We agree with your comment on the limited scope of experiments. To address this, we have conducted additional experiments using Qwen-2.5-7B and Qwen-2.5-32B on both GSM8K and MATH500 datasets. The expanded experimental results are detailed below:
>
> | Model           | GSM8K | MATH500 |
> |-----------------|-------|---------|
> | Qwen-2.5-7B     | 85.5  | 49.2    |
> | + cDPO          | 87.5  | 54.0    |
> | Qwen-2.5-32B    | 93.0  | 58.8    |
> | + cDPO          | 93.5  | 64.8    |
>
> Additionally, we have also evaluated Pass@1 accuracy under various temperature settings. We sampled each question 10 times for each temperature setting and report average Pass@1 accuracy:
>
> | Temperature (T)        | 0    | 0.25 | 0.5  | 0.75 | 1.0  | 1.25 | 1.5  |
> |------------------------|------|------|------|------|------|------|------|
> | Llama-3-8B             | 18.6 | 16.4 | 15.3 | 13.0 | 9.5  | 3.3  | 1.2  |
> | + cDPO                 | 19.6 | 20.3 | 20.1 | 19.6 | 18.7 | 19.7 | 18.3 |
> | DeepSeek-math-7B       | 34.0 | 31.8 | 30.5 | 26.2 | 21.1 | 11.3 | 3.0  |
> | + cDPO                 | 35.2 | 34.5 | 34.5 | 34.5 | 33.9 | 32.9 | 32.8 |
> | Qwen-2.5-7B            | 49.2 | 46.9 | 45.1 | 41.4 | 34.0 | 20.1 | 2.8  |
> | + cDPO                 | 54.0 | 54.2 | 53.6 | 52.8 | 52.9 | 53.4 | 51.9 |
>
> These results demonstrate clearly that:
>
> - cDPO consistently surpasses the baseline model performance.
> - cDPO maintains stability and robustness across diverse temperature settings.
>
> **W3. Ablation of the proposed methods**
>
> Thank you for the valuable suggestion. To strengthen our analysis, we have conducted an ablation study on the hyperparameter β controlling the mean of the contrastive distribution P^{ce}, as explained in the Appendix section “Distribution Analysis of Contrastive Estimation”. Using LLaMA-3-8B on GSM8K, we observed the performance effects of β values, shown below:
>
> | β-value | 0.5  | 0.75 | 1.0  | 1.25 | 1.5  | 1.75 | 2.0  | 2.25 | 2.5  |
> |---------|------|------|------|------|------|------|------|------|------|
> | cDPO    | 66.5 | 69.2 | 67.9 | 67.2 | 69.5 | 70.7 | 68.4 | 66.8 | 68.9 |
>
> These results indicate that optimal performance occurs when β is set within a range of 1.5–1.75, highlighting the need to appropriately balance the amplification and suppression of token likelihoods during training. We will present a more detailed discussion and expanded analysis on this topic in the next revision of the manuscript.

---

> > ### Comment · Reviewer_KGMH · 2025-04-03
> >
> > The rebuttal partly solved my concerns. So I raise my score to weak accept.

---

> > > ### Author Response · Authors · 2025-04-09
> > >
> > > Dear Reviewer KGMH,
> > >
> > > Thank you for your reply and for updating your score! We will incorporate the experimental results presented in the rebuttal into the revised version of the paper. We truly appreciate your time and support in helping us improve our work.
> > >
> > > Best.
> > >
> > > Submission 2521 Authors

---

### Official Review · Reviewer_Qri4 · 2025-03-14

**Overall Recommendation:** 4

**Summary:**

This paper introduces the concept of critical tokens, which are tokens that significantly influence the reasoning trajectories, leading to incorrect outcomes. They propose to use rollout algorithm to identify critical tokens, then study the difference between critical tokens and wrong tokens. They further propose an efficient method to detect the critical tokens. Based on these findings, they propose a cDPO method that assign more weights on critical tokens, and show that it improves the performance.

**Claims And Evidence:**

The claims for critical tokens are convincing: The authors demonstrate the existence of critical tokens. The criteria chosen by the authors are very stringent. Nevertheless, they successfully identify a large amount of critical tokens, showing that critical tokens widely appear in LLM generations.

However, I find the claims for the efficient identification method less convincing. There lacks comparison between the efficient identification approach and the "golden standard", the rollout algorithm. I can only find the efficiency comparison. There should be a table showing the correct identification probability of critical tokens for the efficient algorithm. Even if the two methods do not lead to coherent approach, it will be important to understand the cause of the difference.

**Essential References Not Discussed:**

None

**Experimental Designs Or Analyses:**

1. Can the authors explain more about the derivation of Equation (1)? For example, why does s_t can be realised with this form? How does this form fit into the weights in the proposed cDPO?
2. When is the negative model trained? Is it prior to the cDPO training?
3. There should be more details on the implementation of the rollout method.

**Methods And Evaluation Criteria:**

There are several concerns:
1. The use of base model instead of instruct fine-tuned model: Can the authors comment on the choice of the model? It would make more sense to use the instruct finetuned model as the baseline and for further experiment.
2. There should also be some details for evaluation (COT, k-shot, pass@k, etc).
3. Since the negative model serves only as an intermediate step and is discarded afterward, the current approach introduces significant computational overhead through the negative model used for critical token identification.

[1] L. Team and A. Meta, “The Llama 3 Herd of Models,” Jul. 2024. Available: https://arxiv.org/pdf/2407.21783
‌

‌

**Other Comments Or Suggestions:**

None

**Other Strengths And Weaknesses:**

None.

**Questions For Authors:**

None

**Relation To Broader Scientific Literature:**

The critical tokens fit with previous intuitions that some tokens are more important than others, both in the general setting [1], or in mathematical reasoning setting [2]. This may also relate to COT compression problems.

[1] Lin, Zhenghao, Zhibin Gou, Yeyun Gong, Xiao Liu, Ruochen Xu, Chen Lin, Yujiu Yang, Jian Jiao, Nan Duan, and Weizhu Chen. "Not all tokens are what you need for pretraining." Advances in Neural Information Processing Systems 37 (2024): 29029-29063.

[2] Xia, Heming, Yongqi Li, Chak Tou Leong, Wenjie Wang, and Wenjie Li. "Tokenskip: Controllable chain-of-thought compression in llms." arXiv preprint arXiv:2502.12067 (2025).

**Theoretical Claims:**

I did not check the theoretical claims as I think the paper mostly focus on empirical applications.

---

> ### Author Rebuttal · Authors · 2025-04-01
>
> Thank you for your thoughtful and constructive review. We sincerely appreciate your recognition of our work's novelty and contributions. Your detailed comments are very helpful; we respond to each of your concerns below.
>
> **Concerns:**
>
> **C1. Comparison Between Contrastive Estimation (CE) and the Rollout Algorithm**
>
> The AUC values of CE compared to rollout sampling on LLaMA-3-8B are as follows:
> 1. GSM8K: 0.77
> 2. MATH: 0.84
>
> We will include these results in the next revision.
>
> **C2. Use of Base Model Instead of Instruct-Tuned Model**
>
> We primarily adhere to setting protocols established by previous studies [1, 2, 3] by using base models rather than instruct-tuned models. This choice serves two main purposes:
> 1. To ensure controlled and consistent model evaluations.
> 2. To isolate the effects specific to the post-training phase, thereby clearly assessing our method’s direct contribution without interference from any prior fine-tuning effects.
>
> **C3. Detailed Information on Evaluation Setup (COT, k-shot, pass@k, etc.)**
>
> Thank you for pointing out the need for clarification. Our experimental configurations strictly follow established practices:
> - 8-shot prompting for GSM8K as in [4], and 4-shot prompting for MATH500 as described in [5];
> - Temperature fixed at 0 for main evaluations.
>
> Furthermore, we perform additional analyses by sampling each question 10 times at varying temperatures and measure Pass@1 accuracy as shown below:
>
> | Temperature (T) | 0 | 0.25 | 0.5 | 0.75 | 1.0 | 1.25 | 1.5 |
> |-----------------|---|------|-----|------|-----|------|-----|
> | LLaMA-3-8B      |18.6|16.4  |15.3 |13.0  |9.5  |3.3   |1.2  |
> | + cDPO          |19.6|20.3  |20.1 |19.6  |18.7 |19.7  |18.3 |
> | DeepSeek-math-7B|34.0|31.8  |30.5 |26.2  |21.1 |11.3  |3.0  |
> | + cDPO          |35.2|34.5  |34.5 |34.5  |33.9 |32.9  |32.8 |
> | Qwen-2.5-7B     |49.2|46.9  |45.1 |41.4  |34.0 |20.1  |2.8  |
> | + cDPO          |54.0|54.2  |53.6 |52.8  |52.9 |53.4  |51.9 |
>
> Key observations:
> - cDPO consistently outperforms baseline models across all temperatures.
> - cDPO demonstrates stable and robust performance under varying sampling conditions.
>
> **C4. Computational Overhead of Training the Negative Model**
>
> We appreciate your insightful observation. Identifying critical tokens via rollout sampling incurs significant computational overhead. To address this limitation, we introduce contrastive estimation (CE), a computationally efficient alternative utilizing trained positive and negative models to estimate critical tokens. To validate this approach further, we incorporate CE-based scoring into our cDPO strategy and demonstrate its effectiveness experimentally.
>
> Looking ahead, if sufficiently extensive datasets annotated with critical tokens become available, training a dedicated token-level reward predictor model can deliver an even more scalable and lightweight alternative solution.
>
> ---
>
> **Questions:**
>
> **Q1. Derivation of Equation (1)**
>
> - *Clarification of Derivation*: As thoroughly discussed in the Appendix "Distribution Analysis of Contrastive Estimation," the term s_t presented in Equation (1) is derived directly as a combination of the probability distributions P^p (positive model) and P^n (negative model). This formulation results in an improved distribution, efficiently suppressing incorrect token likelihoods while promoting correct ones.
> - *Use of Logits as Weights in cDPO*: Further, as detailed in Section 3.2 "Formulation", cDPO refines the negative portion of the DPO loss into a token-level loss, using s_t as weighting factors. This mechanism explicitly guides training towards avoiding generation of critical tokens.
>
> **Q2. Timing for Training the Negative Model**
>
> As depicted explicitly in Figure 3, the negative model is trained prior to initiating cDPO training. Specifically, our full experimental pipeline consists of the following two distinct phases:
> 1. Critical token estimation using CE, involving prior training of both positive and negative models.
> 2. Integration of CE-derived scores into cDPO training.
>
> **Q3. Implementation Details of the Rollout Method**
>
> We perform rollout sampling exactly as described in Lines 194–202:
> - Given an incorrect response T = {t_1, t_2, ..., t_n}, we traverse each token t_i.
> - At each position t_i, we generate k=64 sampled continuations from the prefix T_{≤i} , employing the identical sampling configuration used for original generation.
> - The accuracy averaged over these k continuations forms a score for token t_i.
>
> ---
>
> **References:**
>
> [1] ARGS: Alignment as Reward-Guided Search. ICLR 2024.
>
>  [2] Token-level direct preference optimization. ICML 2024.
>
>  [3] Enhancing llm reasoning via critique models with test-time and training-time supervision. arXiv 2024.
>
>  [4] Chain-of-thought prompting elicits reasoning in large language models. NeurIPS 2022.
>
>  [5] Solving quantitative reasoning problems with language models. NeurIPS 2022.

---

### Decision · Program_Chairs · 2025-05-01

**Decision:**

Accept (poster)

**Comment:**

The paper is well-written and easy to follow. The authors propose a simple yet reasonable definition of critical tokens and successfully identify a large number of them in their experiments, demonstrating that critical tokens are prevalent in LLM outputs. They also leverage this method to stabilize DPO training, which adds practical value. Overall, this is a solid piece of work, and all reviewers are in favor of acceptance.